# Improved baselines for vision-language pre-training

**Enrico Fini**[*1,2△], **Pietro Astolfi**[*1▽], **Adriana Romero-Soriano**[§1,3,4,5],
**Jakob Verbeek**[§1], **Michal Drozdzal**[§1]

[1]*FAIR, Meta,* [2]*University of Trento,* [3]*Mila, Quebec AI Institute,* [4]*McGill University,* [5]*Canada CIFAR AI Chair*
[△]*enrico.fini@gmail.com,* [▽]*pietroastolfi@meta.com*
[*,§]*Contributed equally*

**Reviewed on OpenReview:** *https://openreview.net/forum?id=a7nvXxNmdV*

## Abstract

Contrastive learning has emerged as an efficient framework to learn multimodal representations. CLIP, a seminal work in this area, achieved impressive results by training on paired image-text data using the contrastive loss. Recent work claims improvements over CLIP using additional non-contrastive losses inspired from self-supervised learning. However, it is sometimes hard to disentangle the contribution of these additional losses from other implementation details, *e.g.*, data augmentation or regularization techniques, used to train the model. To shed light on this matter, in this paper, we first propose, implement and evaluate several baselines obtained by combining contrastive learning with recent advances in self-supervised learning. In particular, we use the loss functions that were proven successful for visual self-supervised learning to align image and text modalities. We find that these baselines outperform a basic implementation of CLIP. However, when a stronger training recipe is employed, the advantage disappears. Indeed, we find that a simple CLIP baseline can also be improved substantially, up to a 25% relative improvement on downstream zero-shot tasks, by using well-known training techniques that are popular in other subfields. Moreover, we discover that it is enough to apply image and text augmentations to make up for most of the improvement attained by prior works. With our improved training recipe for CLIP, we obtain state-of-the-art performance on four standard datasets, and consistently outperform prior work (up to +4% on the largest dataset), while being substantially simpler. The code is available at https://github.com/facebookresearch/clip-rocket

## 1 Introduction

Vision-language pre-training (VLP) is a recent learning paradigm that enables neural networks to learn multimodal representations from images and text. The learned representations are inherently semantic and can be easily transferred to several tasks, such as zero-shot image classification and multimodal retrieval. Arguably, the most prominent approach to VLP is CLIP (Radford et al., 2021), which leverages a contrastive loss to learn aligned image and text representations from large-scale image-text datasets scraped from the internet.

Notwithstanding its success, CLIP is not without its limitations. These include a high computational cost and a strong sensitivity to the size and composition of data batches used during training due to the contrastive approach, as well as other limitations such as a lack of fine-grained alignment between image regions and words in a sentence, and the absence of cross-modality feature fusion. Moreover, the standard implementation of CLIP adopts a rather basic training recipe, which does not incorporate commonly used training techniques such as image and text augmentations (Shorten & Khoshgoftaar, 2019), label smoothing (Müller et al., 2019), and architectural improvements like projector networks (Chen et al., 2020; Grill et al., 2020), which have been shown to enhance generalization and robustness in supervised and self-supervised representation learning (SSL).

Motivated by these limitations, recent VLP research has sought to improve upon CLIP by modifying the contrastive objective and integrating advanced training techniques. In particular, to improve the contrastive objective, some works (Yao et al., 2021; Gao et al., 2022; Fürst et al., 2022; You et al., 2022) have tried to tackle the fine-grained image-text alignment or cross-modal feature fusion. Others, *e.g.*, SLIP (Mu et al., 2022) and DeCLIP (Li et al., 2021), have regularized the objective with auxiliary self-supervised image-to-image losses. However, the contrastive approach remains a limiting factor. For this reason, some researchers have suggested using vision-language non-contrastive losses (Chen et al., 2022; Zhou et al., 2022; Singh et al., 2022). These losses are reminiscent of SSL, where they have been theoretically well-studied (Tian et al., 2021; Balestriero & LeCun, 2022), and their translation to VLP has the potential to solve the challenge of the batch size sensitivity and heavily reduce the computational cost. However, such existing methods do not provide faithful VLP translations of SSL losses and have not explored many well-known non-contrastive losses successful in visual SSL, *e.g.*, SwAV (Caron et al., 2020), BYOL (Grill et al., 2020), SimSiam (Chen & He, 2021), Barlow Twins (Zbontar et al., 2021). Furthermore, as in many CLIP extensions, the dual adoption of modified objectives and advanced training techniques, which despite re-implementations (Ilharco et al., 2021; Li et al., 2021) are not employed by baseline CLIP, makes it hard to evaluate the impact of such methods. It emerges that the VLP literature is far from standardizing the training recipe and would benefit from the analysis of advanced training recipes to clarify the landscape of the state-of-the-art.

In this paper, we aim to elucidate the state of the art for VLP by systematically investigating non-contrastive extensions of CLIP, and tuning an *improved* training recipe for CLIP that can reveal the actual impact of ours as well as existing CLIP extensions. In particular, we present four new VLP baselines: SiamLIP, BYOLIP, BarLIP, and SwALIP, by translating the non-constrastive losses of the SSL methods, SimSiam, BYOL, Barlow Twins, and SwAV to the multimodal case. We test these methods in the zero-shot image classification task and we find that they cannot learn strong aligned multimodal features on their own (as also found by concurrent work (Zhou et al., 2022)), but can be used in composition with the CLIP loss to boost the performance compared to a standard CLIP baseline. However, we also find that this improvement is much smaller than the increase in performance brought about by advanced training techniques. Indeed, we carefully explore training recipes for VLP, by combining and evaluating the training techniques that are best practices in the supervised (Dosovitskiy et al., 2020; Wightman et al., 2021) and self-supervised (Chen et al., 2020) literature. This results in the proposal of an improved training recipe, dubbed 🚀, that comprises image and text augmentations (Shorten & Khoshgftaar, 2019; Shorten et al., 2021), dropout (Srivastava et al., 2014), and label smoothing (Müller et al., 2019), combined with architectural design choices (*i.e.* design of the projection layers). By pre-training on four public datasets of various sizes, we discover that when employed for CLIP (CLIP 🚀), this recipe substantially improves its zero-shot ImageNet performance (up to 11%), so much that it surpasses of up to 4% all previously published results obtained with more complex CLIP extensions, including our non-contrastive baselines also trained with the same improved recipe. We conclude that, although methodologically interesting, these more complicated models do not provide enhanced results. In practice it is better to stick to a well-tuned implementation of CLIP than rely on existing additional losses.

Our contributions can be summarized as follows:

- We systematically analyze non-contrastive losses for vision-language pre-training by proposing four baselines that outperform a standard implementation of CLIP by a few points (1-2%) on ImageNet;

- We collect and study advanced training recipes for CLIP, and propose a well-tuned improved recipe, CLIP 🚀, that dramatically boosts its performance up to 11% on ImageNet (relative boost ∼25%);

- We reset the state of the art for VLP by showing that CLIP 🚀 is better than methods employing additional or modified contrastive and non-contrastive losses when pre-trained on four datasets, ranging from 3M to 29M image-text pairs. In particular, using the largest dataset we outperform by 4% the best related method.

## 2 Background

Let us consider a dataset $\mathcal{D} = \{(\boldsymbol{x}_i, \boldsymbol{y}_i)\}_{i=1}^M$ of $M$ paired data points $\boldsymbol{x}_i$ and $\boldsymbol{y}_i$ representing the same underlying concept. In our manuscript, $\boldsymbol{x}_i$ represents an image and $\boldsymbol{y}_i$ represents either a data augmentation

or textual description of $\boldsymbol{x}_i$. Moreover, let us define two encoding functions $A$ and $B$ that project input data into representations: $\boldsymbol{z}_i^A = A(\boldsymbol{x}_i)$ and $\boldsymbol{z}_i^B = B(\boldsymbol{y}_i)$. Note that if $\boldsymbol{x}_i$ and $\boldsymbol{y}_i$ are from the same modality (*e.g.*, images), $A$ and $B$ may represent the same encoding. The objective of representation learning is to learn robust encoding function that produces representations that generalize well to diverse downstream tasks.

## 2.1 Contrastive learning

In contrastive learning, the main idea is to minimize the distance between different representations $\boldsymbol{z}_i^A$ and $\boldsymbol{z}_i^B$ of the same data pair $(\boldsymbol{x}_i, \boldsymbol{y}_i)$ – *positive* pairs – and maximize the distance between representations $\boldsymbol{z}_i^A$ and $\boldsymbol{z}_j^B$ from different data pairs $(\boldsymbol{x}_i, \boldsymbol{y}_i)$ and $(\boldsymbol{x}_j, \boldsymbol{y}_j)$ – *negative* pairs. Formally, this is achieved by optimizing the following objective:

$$\mathcal{L}_{\mathrm{C}}^{A \to B} = -\sum_{i=1}^{N} \log \frac{\exp\left(\mathrm{sim}\left(\boldsymbol{z}_i^A, \boldsymbol{z}_i^B\right)/\tau\right)}{\sum_{j=1}^{N} \exp\left(\mathrm{sim}\left(\boldsymbol{z}_i^A, \boldsymbol{z}_j^B\right)/\tau\right)}, \tag{1}$$

where the summation is over $N$ elements in the batch, $\tau$ is the temperature hyper-parameter and $\mathrm{sim}(\cdot, \cdot)$ is the cosine similarity between two vectors. Note that if $A$ and $B$ are different, the loss is not symmetric; we use the superscript $A \to B$, to denote the direction of similarity computation. Nevertheless, the contrastive loss can be made symmetric by considering both similarity computation directions:

$$\mathcal{L}_{\mathrm{C}} = \mathcal{L}_{\mathrm{C}}^{A \to B} + \mathcal{L}_{\mathrm{C}}^{B \to A}. \tag{2}$$

The contrastive objective in Eq. 1 is used both in SSL – where $\boldsymbol{y}_i$ represents an augmented version of $\boldsymbol{x}_i$ – and VLP – where $\boldsymbol{y}_i$ represents the text description associated with $\boldsymbol{x}_i$. CLIP (Radford et al., 2021) is a *de facto* standard for VLP which leverages cross-modal contrastive learning. In practice, the model leverages a vision encoder and a text encoder to obtain image and text representations, respectively. In the typical implementations, the vision encoder is either a ResNet (He et al., 2016) or a vision transformer (Dosovitskiy et al., 2020) while the text encoder is a text transformer that processes the input text to produce a fixed-length vector representation.

**Limitations of contrastive learning.** One disadvantage of the contrastive learning in CLIP is that the quality and diversity of the learned features can be affected by the choice of negative samples (Chen et al., 2020; He et al., 2020). If the negative samples are not diverse enough, the model may learn to distinguish between only a small subset of dissimilar samples, leading to poor performance. For this reason, the contrastive loss is usually optimized using large batch sizes, which are more likely to carry larger sample diversity (Pham et al., 2021; Zhai et al., 2023). However, increasing the batch size also increases the memory footprint of the model. This issue is exacerbated by the fact that the size of the similarity matrix scales quadratically with the batch size. Another disadvantage of the contrastive loss is that it may not be tolerant to semantically similar samples (Zhong et al., 2021). For instance, if the batch contains many instances of the same concept it may map them far apart in the feature space, breaking the underlying semantic structure and harming the emergence of features that are useful for downstream tasks. These limitations apply to both contrastive SSL and VLP approaches, including CLIP. However, in practice, the second limitation may be mitigated when training on very broad data distributions, as typical for CLIP, because it is less likely to simultaneously sample redundant concepts. However, CLIP is not exempt from this issue when trained on small datasets, and it is worth mentioning that in a recent study, Abbas et al. (2023) observe a high degree of redundancy also in well-known large-scale datasets such as LAION (Schuhmann et al., 2022).

## 2.2 Non-contrastive learning

To overcome the limitations of contrastive learning discussed in Sec. 2.1, non-contrastive learning approaches have been introduced in the literature. In particular, non-contrastive techniques have emerged and been widely explored in self-supervised visual representation learning, where they have repeatedly shown to be effective. These approaches do not rely on the use of negative samples during training, but instead impose consistency among positive pairs (Chen & He, 2021; Grill et al., 2020), replace negatives with a redundancy reduction terms (Zbontar et al., 2021; Bardes et al., 2021) or cluster prototypes (Caron et al., 2018; 2021).

**Consistency-based methods.** Consistency-based methods, such as BYOL (Grill et al., 2020) and Sim-Siam (Chen & He, 2021), optimize consistency objectives like mean squared error (MSE) or negative cosine similarity loss only between positive samples, therefore avoiding the use of negative samples in the loss. This not only reduces the memory requirements during training – as the similarity between all pairs of samples in a batch no longer needs to be computed –, but also results in better performance when training with small batch sizes. The loss function of these methods can be written as follows:

$$\mathcal{L}_{\text{MSE}}^{A \to B} = -\sum_{i=1}^{N} \|\boldsymbol{q}_i^A - \boldsymbol{z}_i^B\|_2^2, \tag{3}$$

where $\boldsymbol{q}_i^A$ is the output of a predictor network that takes as an input $\boldsymbol{z}_i^A$. The encoder $B$ is either a momentum encoder in the case of BYOL or the same encoder as $A$ in the case of SimSiam. In principle, this loss function is commutative, however, since these methods exhibit an asymmetric architecture – due to the presence of the predictor – the loss can be made symmetric by swapping the encoders $A$ and $B$ and attaching the predictor network to the output of the encoder $B$, similar to Eq. 2.

**Redundancy reduction methods.** Redundancy reduction methods represent another example of methods using a non-contrastive loss in visual SSL. We identify three methods that belong to this category: Barlow Twins (Zbontar et al., 2021), VICReg (Bardes et al., 2021) and W-MSE (Ermolov et al., 2021). For simplicity, here we describe the mechanism and the loss function in Barlow Twins, the most influential of these three methods, but similar arguments can be made for all of them.

Barlow Twins (Zbontar et al., 2021) uses a cross-correlation loss that decorrelates feature components and maximizes variability of learned representations. More precisely, the loss measures the cross-correlation matrix between the outputs of two identical networks fed with distorted versions of a sample and tries to make it as close to the identity matrix as possible. This causes the representations of distorted versions of a sample to be similar while minimizing redundancy between the components of these representations. The loss function reads as:

$$\mathcal{L}_{\text{XC}} = \sum_{u} \left(1 - \mathcal{C}_{uu}\right)^2 + \lambda \sum_{u} \sum_{v \neq u} \mathcal{C}_{uv}^2, \tag{4}$$

where $\lambda$ is a weight and

$$\mathcal{C}_{uv} \triangleq \frac{\sum_{i=1}^{N} \boldsymbol{z}_{i,u}^A \boldsymbol{z}_{i,v}^B}{\sqrt{\sum_{i=1}^{N} \left(\boldsymbol{z}_{i,u}^A\right)^2} \sqrt{\sum_{i=1}^{N} \left(\boldsymbol{z}_{i,v}^B\right)^2}}$$

is the cross-correlation matrix between $\boldsymbol{z}_i^A$ and $\boldsymbol{z}_i^B$ along the batch dimension.[1] In the case of Barlow Twins, $A$ and $B$ represent the same encoding function.

**Clustering-based methods.** The third and last family of visual SSL methods is based on clustering. We identify three methods that belong to this category: SwAV (Caron et al., 2020), DINO (Caron et al., 2021), and DeepClusterV2 (Caron et al., 2018). By means of clustering, these methods discretize the feature space to learn representations, which enable them to use the cross-entropy loss to compare cluster assignments of different views of the same sample, as follows:

$$\mathcal{L}_{\text{XE}}^{A \to B} = -\sum_{i=1}^{N} \sum_{k=1}^{K} \boldsymbol{a}_{i,k}^B \log \frac{\exp\left(\text{sim}\left(\boldsymbol{z}_i^A, \boldsymbol{p}_k\right)/\tau\right)}{\sum_{k'=1}^{K} \exp\left(\text{sim}\left(\boldsymbol{z}_i^A, \boldsymbol{p}_{k'}\right)/\tau\right)}, \tag{5}$$

where $\boldsymbol{p}_k$ is the $k$-th learnable cluster prototype, $\boldsymbol{a}_i^B$ is the vector of cluster assignments of $\boldsymbol{z}_i^B$, and $K$ is the number of clusters. In the case of SwAV, $A$ and $B$ represent the same encoding function, while in DINO $B$ is a momentum encoder. Similarly to consistency-based methods, this cross-entropy-based loss is asymmetric, but can be made symmetric by swapping $A$ and $B$. In the remainder of the manuscript we focus on SwAV. In SwAV, assignments are obtained by using the Sinkhorn-Knopp algorithm (Cuturi, 2013), which approximates optimal transport by iteratively normalizing the rows and the column of the matrix containing the similarities of samples in the batch, $\boldsymbol{z}_{i,\forall i \in \{1,...,N\}}^B$ and the prototypes $\boldsymbol{p}_{k,\forall k \in \{1,...,K\}}$. This equates to imposing that the clusters are roughly equally represented in the batch.

---

[1] We use the double subscript notation $z_{i,j}$ to indicate $j$-th element of the vector $\boldsymbol{z}_i$.

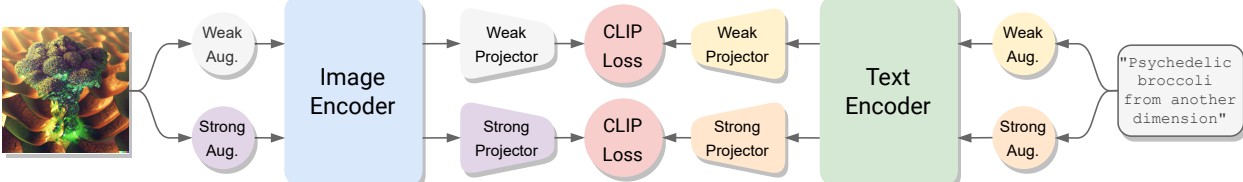

Figure 1: Overall architecture of our CLIP 🚀 model.

## 3   Improved training recipe

Our improved training recipe, which we denote with 🚀, is based on the original CLIP training recipe but incorporates the following important modifications: we incorporate stronger data augmentations, introduce new projector, add label smoothing and additional regularization for the text encoder.

**Image augmentations.**   In CLIP, the only augmentation, *i.e.* weak augmentation, is obtained by randomly cropping a large portion of the input image. Instead, in our improved recipe we generate additional views of the same input image by applying multiple image transformations that produce strong augmentations. Specifically, inspired by the best practices in supervised (see, *e.g.*, Wightman et al. (2021)) and self-supervised learning (see, *e.g.*, Chen et al. (2020)), we first crop the image by randomly sampling the crop scale from a wider range than the one used in the weak augmentation; then, we sequentially apply a random combination of color distortions such as contrast, brightness and saturation jittering and color dropping (greyscaling), Gaussian blurring, and horizontal flipping.

**Text augmentations.**   We borrow text augmentations from the NLP literature, where a variety of transformations that do not alter semantic have been presented, see, *e.g.*, Shorten et al. (2021). We generate the weak augmentation by simply removing stop-words with an arbitrary probability. Note that this transformation has been already shown to be useful for vision-language pre-training by Tejankar et al. (2021). Moreover, to generate stronger augmentations, if multiple captions are available for the same image, we independently and uniformly sample one caption among them for each augmentation, otherwise, we use the same caption for all the augmentations. Then, we combine stop-word removal with a one randomly sampled transformation among synonym replacement, word insertion/deletion, and sentence shuffling, as suggested by Wei & Zou (2019). Note that the original CLIP training recipe does not exploit any text augmentations.

**Projector design.**   CLIP employs linear projectors to embed text and image representations into the same space. However, in the vision literature multi-layer projectors have emerged as an indispensable element when employed in combination with strong data augmentation. This fact has been empirically proved in multiple learning scenario, *e.g.* supervised (Khosla et al., 2020; Sariyildiz et al., 2023), semi-supervised (Fini et al., 2021; 2023; Assran et al., 2021), and self-supervised (Caron et al., 2021), and an intuition of the explanation is provided by Bordes et al. (2021). Grounded by this, we adopt multi-layer perceptrons (MLP), consisting of two linear layers separated by a BatchNorm and a ReLu layer as *strong projectors* for both image and text, while we keep linear projectors as in original CLIP to embed weak image and text augmentations.

**Label smoothing.**   Label smoothing (Szegedy et al., 2016) discourages the model from being overconfident in its predictions. This is especially helpful when the dataset is noisy or the data augmentation recipe is so strong that it can change the semantics of the input, or the model overfits the training distribution. In CLIP 🚀, since we have both strong augmentations and deep projectors, we propose to soften the identity matrix otherwise used as target in the contrastive loss (see Sec. 2.1). With the same logic, we do not apply label smoothing when computing the contrastive loss between weak image and text augmentations embedded with linear projectors.

**Overview of our recipe** An overview of CLIP 🚀 approach is depicted in Fig. 1 and in the pseudo-code of Pseudo-code 1 (see appendix for the exact implementation). At each training step, we load a batch of text-image pairs. Each pair is augmented creating one weak and one or more strong augmentations of a pair [2]. The augmented images and texts are then fed to the image and text encoders, respectively. Importantly, we apply dropout to the text encoder (transformer) to regularize it. Then, the embedded weak and strong augmentations are projected into the shared cross-modal space using two separate projectors (weak and strong projector). The model is trained with the standard CLIP loss (see Eq. 2) applied to the corresponding views of image and text, *i.e.* projected weak image augmentations are contrasted to projected weak text augmentation, and vice-versa for projected strong image and text augmentations. When computing the loss between strong augmentations we apply label smoothing. At inference time, we average the similarities predicted by the two projectors (weak and strong) to obtain the final similarity score between image and text.

---

**Pseudo-code 1** CLIP 🚀 training procedure

```
# This code describes the case of 1 weak and 1 strong aug
# x, y: image and its corresponding text
# tau_weak, tau_strong: learnable temperatures
# label_smooth: label smoothing parameter
# image_enc, text_enc: encoders A & B (possibly w/ dropout)
def training_step(x, y):

    # weak augmentations
    x_weak = image_augment(x, mode="weak")
    y_weak = text_augment(y, mode="weak")

    # strong augmentations
    x_strong = image_augment(x, mode="strong")
    y_strong = text_augment(y, mode="strong")

    # forward weak
    z_image_weak = image_proj_weak(image_enc(x_weak))
    z_text_weak = text_proj_weak(text_enc(y_weak))

    # forward strong
    z_image_strong = image_proj_strong(image_enc(x_strong))
    z_text_strong = text_proj_strong(text_enc(y_strong))

    # compute loss
    loss_weak = contrastive_loss(
        z_image_weak,
        z_text_weak,
        temperature=tau_weak
    )
    loss_strong = contrastive_loss(
        z_image_strong,
        z_text_strong,
        temperature=tau_strong,
        label_smooth=label_smooth
    )
    return loss_weak + loss_strong
```

---

## 4 Non-contrastive baselines for vision-language pre-training

In Sec. 2 we reviewed the use of non-contrastive losses in SSL. Here, we propose to incorporate such losses into the vision-language pre-training of CLIP, with the idea of mitigating the issues of the CLIP contrastive training objective. In our non-contrastive baselines, we consider the following training objective:

$$\mathcal{L} = \alpha \mathcal{L}_{\text{C}} + \beta \mathcal{L}_{\text{NC}}, \tag{6}$$

where $\mathcal{L}_{\text{C}}$ is the contrastive loss of CLIP, $\mathcal{L}_{\text{NC}}$ is a non-contrastive auxiliary loss and $\alpha, \beta \in [0, 1]$ are weights. Note that both $\mathcal{L}_{\text{C}}$ and $\mathcal{L}_{\text{NC}}$ are intended as cross-modals between image and text. Note that, setting $\alpha = 1$ and $\beta = 0$ is equivalent to using only CLIP loss only, while $\alpha = 0$ and $\beta = 1$ results in non-contrastive multi-modal learning. In our experiments, we found that setting $\alpha = \beta = 1$ worked best. See supplementary for the full ablation.

In this section, we consider four types of non-contrastive losses and combine them with CLIP to build four models: SiamLIP, BYOLIP, BarLIP, and SwaVLIP. Unless otherwise specified, all the baselines use a multi-layer projector to refine the features.

**SiamLIP & BYOLIP.** SiamLIP & BYOLIP (see Fig. 2a) are two VLP models that complement the contrastive training objective of CLIP with a consistency-based objective. In the case of SiamLIP & BYOLIP, the complementary training objective is inspired by SimSiam (Chen & He, 2021) and BYOL (Grill et al., 2020). For these models, we define $\mathcal{L}_{\text{NC}}$ to be the cross-modal version of $\mathcal{L}_{\text{MSE}}$ (see Eq. 3), computed in both directions $A \rightarrow B$ and $B \rightarrow A$, where $A$ is the image encoder and $B$ is the text encoder. The use of an asymmetric training objective requires the architecture to have two projectors followed by two predictors, mapping from $A \rightarrow B$ and $B \rightarrow A$, respectively. Moreover, following BYOL, in BYOLIP we use momentum encoders for both image and text modalities.

---

[2]Throughout the manuscript, we refer to original CLIP augmentations as *weak* augmentations.

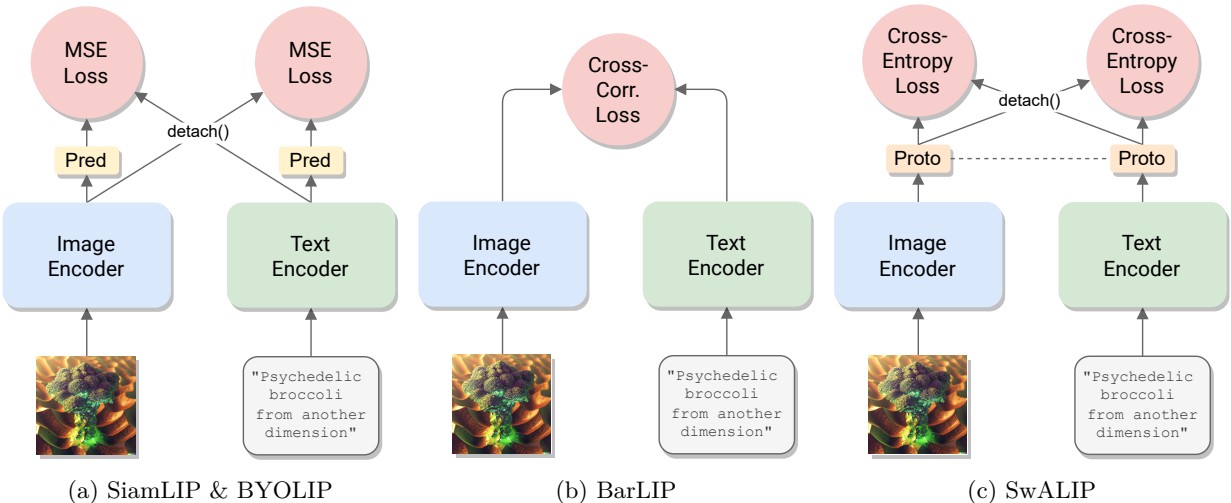

(a) SiamLIP & BYOLIP       (b) BarLIP       (c) SwALIP

Figure 2: Overview of the proposed improved baselines: SiamLIP, BYOLIP, BarLIP and SwALIP. For the sake of visualization, we do not show the CLIP loss used in combination to the non-contrastive loss and the projector networks in all the models. For BYOLIP we do not show the additional momentum encoder.

**BarLIP.** BarLIP (see Fig. 2b) is a VLP model that combines CLIP objective with a redundancy reduction criteria. In particular, we take the inspiration from Barlow Twins (Zbontar et al., 2021) and adapt Eq. 4 to the VLP scenario and use it as the non-contrastive loss ($\mathcal{L}_{\mathrm{NC}}$). Contrary to Barlow Twins, in BarLIP we use two encoders – $A$ for vision and $B$ for text inputs. The multimodal version of Eq. 4 aligns the feature space of both data modalities while also decorrelating its components, which should help when pre-training on noisy and duplicate-rich datasets typical of vision-language pre-training.

**SwALIP.** SwALIP (see Fig. 2c) is a VLP model that combines CLIP objective with clustering-based criteria. In particular, we get inspiration from SwAV (Caron et al., 2020) and introduce the cross-modal adaptation of the cross-entropy loss, $\mathcal{L}_{\mathrm{XE}}$ (see Eq. 5), to compare the cluster assignments of different representations of the same concept. In this case, the encoder $A$ takes as an input images and the encoder $B$ works on texts. The learned set of prototypes are shared across the two modalities, and basically partitions concepts in both modalities in the aligned feature space. The cross-entropy loss is computed in both directions enforcing that paired image and text samples belong to the same cluster. However, in practice, this formulation for SwALIP turns out to be quite unstable, and for most our experiments we adopt a slightly modified version of SwALIP that does not exhibit learned prototypes, rather, it uses representations from the text modality as prototypes. Note that this modified formulation computes the swapped assignments using correlated views of each image-text pair; hence, it requires multiple views as in our improved training recipe, and cannot be used otherwise. We refer to the appendix for additional details and the formalization of modified SwALIP.

## 5 Experiments

### 5.1 Datasets and evaluation

We pre-train the presented CLIP variants on the datasets described below, ranging in size from 3M to 29M. We test the models in the zero-shot image classification task, which is performed by computing the cosine similarity between the image representation and the representation of all the classes encoded as text and choosing the most similar class. The performance is measured in terms of accuracy on the ImageNet-1000 (Deng et al., 2009) validation set. Moreover, following Radford et al. (2021), we investigate the models' generalization using an extended set of 22 vision benchmarks of different kinds – most of them belong to the widely adopted Visual Task Adaption Benchmark (VTAB) (Zhai et al., 2019).

The **Conceptual Captions** dataset is composed of image-caption pairs that have been filtered based on: (i) image properties and content; (ii) caption language, information richness, and content; (iii) alignment between image and caption; and (iv) hypernymization and text refinement. Given this pipeline, two datasets have been published in the literature: **CC3M** (Sharma et al., 2018) composed of 3.3M image-text pairs obtained by applying the full filtering pipeline, and **CC12M** (Changpinyo et al., 2021) comprising 12.4M pairs obtained from a relaxed version of the pipeline (relaxed uni-modal filters (i), (ii), and (iv)).

The **Yahoo Flickr Creative Commons** dataset is composed of 100M of image-text pairs (Thomee et al., 2016). Noisy text-image pairs have been filtered by Radford et al. (2021) to obtain a cleaner version of 15M pairs. This dataset is referred to as **YFCC15M**.

To experiment with larger scale datasets we follow Li et al. (2021) and create a dataset as the union of CC3M, CC12M, and YFCC15M, obtaining 29M image-text pairs. We refer to this dataset as **Open29M**. Moreover, we further scale the dataset size by adopting **PMD46M** a 46M subset[3] of the PMD70M dataset (Singh et al., 2022) containing image-text pairs gathered from different data sources including the whole Open29M.

## 5.2 Implementation details

We train models using both the base recipe used by CLIP and our improved training recipe. We provide further implementation details below.

**Architecture.** We focus our main analysis on CLIP variants using ResNet-50 (He et al., 2016) as the vision backbone. However, we also investigate vision transformers (Dosovitskiy et al., 2020) like ViT-S/16 and ViT-B/16. The text encoder is kept as in the original CLIP, except that we change its number of layers based on the dataset size, *e.g.*, for smaller scale dataset like CC3M we use a 6-layer transformer, while for larger datasets we use 12 layers. Moreover, the strong text and vision projectors are two separate MLPs with BatchNorm and ReLU, each having a hidden and output dimension of 4096 and 256, respectively.

**Optimization.** In most experiments, we pre-train the model for 32 epochs following Li et al. (2021), using the AdamW optimizer (Loshchilov & Hutter, 2017) (betas 0.9 and 0.98), with learning rate 0.003 (or 0.002 for experiments on the 29M dataset) regulated by linear warmup (1 epoch) plus cosine scheduler (final learning rate $10^{-5}$). Mini-batches are composed of 4096 image-text pairs. To provide regularization to the training, weight decay is applied with magnitude 0.1 on all parameters except for biases and normalization layers. For the smaller CC3M dataset, we use weight decay 0.5. The dropout probability in text encoder varies depending on the dataset size, *e.g.*, no dropout on YFCC15M and probability 0.2 on CC3M, while label smoothing is applied with a smoothing factor of 0.1 regardless of the dataset.

**Data augmentation.** Both images and text are augmented to produce one weak and two strong augmentations. For the sake of clarity, we report the exact configuration adopted in Pseudo-code 2.

**Pseudo-code 2** Config for image and text augmentations

```
# x: image                                          # y: list of captions
# c, b, s, h: constrast, brightness, saturation, hue   def text_augment(y, mode="weak"):
def image_augment(x, mode="weak"):
                                                           # random multi-caption choice
    if mode == "weak":                                     y = choice(y)
        transform =
            RandomResizedCrop(size=224, scale=(0.5, 1))    # remove stop-words with probability
                                                           y = rm_stopwords(y, prob=0.8)
    elif mode == "strong":
        transform = Compose([                              if mode == "weak": return y
            RandomResizedCrop(size=224, scale=(0.08, 1)),
            RandomColorJitter(                             # draw one EDA augmentation
                c=0.4, b=0.4, s=0.4, h=0.1, prob=0.8),     transform = choice([
            RandomGrayscale(prob=0.2),                         synonym_replacement,
            RandomGaussianBlur(sigma=(0.1, 2), prob=0.5),      random_swap,
            RandomHorizontalFlip(prob=0.5),                    random_deletion
        ])                                                 ], prob=[0.4, 0.4, 0.2])
    return transform(x)                                    return transform(y)
```

Note that strong image augmentations are SimCLR-like (Chen et al., 2020). Stopword removal is set to 0.9 for YFCC15M experiments, and EDA refers to "easy data augmentation" (Wei & Zou, 2019).

---

[3]We use a smaller subset of PMD70M as we were not able to reproduce part of the dataset.

Table 1: Imagenet zero-shot classification accuracy obtained using CLIP in combinations with other non-contrastive losses. Note that all these runs do not use multi-caption and results for CLIP are obtained using our implementation. *: This score differs from the one in Tab. 2, because here multi-caption is not used.

(a) Base vs. improved recipe on YFCC15M

| Method | Base | | Improved 🚀 | |
|--------|------|------|------|------|
| | ViT-S/16 | ResNet-50 | ViT-S/16 | ResNet-50 |
| CLIP | 32.4 | 32.6 | **43.1** | **43.2**$^*$ |
| BarLIP | **34.4** | **32.8** | 42.5 | 42.6 |
| SiamLIP | 33.4 | 32.4 | 42.7 | 42.8 |
| BYOLIP | 33.8 | 32.7 | 42.9 | 42.8 |
| SwALIP | 32.9 | 32.2 | 41.8 | 42.6 |

(b) Scaling pre-training data with ResNet-50

| Method | Improved 🚀 | | |
|--------|------|------|------|
| | CC3M | CC12M | YFCC15M |
| CLIP | 27.4 | **44.4** | **43.2**$^*$ |
| BarLIP | 28.0 | **44.4** | 42.6 |
| SiamLIP | 27.9 | 43.4 | 42.8 |
| BYOLIP | **28.2** | 43.9 | 42.8 |
| SwALIP | 27.8 | 43.7 | 42.6 |

## 5.3 Experimental results

In this section, we report on our experiments to disentangle the use of advanced data-augmentation techniques from the introduction of alternative losses for vision-language pre-training. In Sec. 5.3.1, we report on our main experiments that compare CLIP with variants that include additional non-contrastive SSL-inspired losses (BarLIP, SiamLIP, BYOLIP, SwALIP), and consider them using the original training recipe and our improved recipe. We then compare CLIP with our improved training recipe with state-of-art results from the literature in Sec. 5.3.2. Finally, we ablate the different components of the improved recipe in Sec. 5.3.3.

### 5.3.1 Comparison with baselines

We start by evaluating the impact of non-contrastive losses on cross-modal learning. In Tab. 1a we report the results obtained with the base training recipe and our improved training recipe (marked with 🚀) using ViT-S/16 and ResNet-50 as vision backbones. From the base recipe columns, it can be observed that when employing the ViT backbone, BarLIP, SiamLIP, BYOLIP, and SwALIP all outperform the original CLIP with a gap ranging between 0.5% and 2.4%. However, they perform on par or slightly worse (up to -0.5%) when employing the ResNet backbone, presumably due to the stronger inductive bias of convolutions. Then, looking at the improved recipe columns, our recipe provides a great boost to all methods, and especially to original CLIP (+10.7% and 10.5% with ViT-S and ResNet-50, respectively); +8.1—9.3% and +9.8—10.4% for the improved non-contrastive CLIP variants with ViT and ResNet-50, respectively. Note that, the results for SwALIP with our improved recipe utilize the modified version of SwALIP described in the appendix, which requires multiple views to optimize its additional clustering-based loss. Moreover, using the improved training recipe, BarLIP, SiamLIP, BYOLIP, and SwALIP all perform similarly (the accuracy is in the ranges 41.8—42.9% and 42.6—42.8% with ViT and ResNet-50, respectively), and *worse* than CLIP 🚀 (43.1% and 43.2%) – additional zero-shot evaluations are reported in the appendix. To validate this result we ran experiments on CC3M and CC12M, which contains different amount of data and are known to be less noisy than YFCC15M, see Tab. 1b. To limit the use of computational resources, we ran this investigation only using ResNet-50. From the results, we can notice comparable performance across all the methods, with CLIP 🚀 obtaining slightly worse results with CC3M pre-training (-0.8%), the best results (on-par with BarLIP) with CC12M, and the best results on YFCC15M.

Taken together, these results clearly demonstrate the importance of a strong training recipe relative to combining contrastive and non-contrastive loss functions. This finding is in line with concurrent work investigating a combination of CLIP with MAE (Weers et al., 2023). In the following, we focus on the best performing CLIP 🚀, compare it to the state-of-the-art, and ablate the components of our improved training recipe.

### 5.3.2 Comparison with the state-of-the-art

Given the conspicuous boost of CLIP 🚀 compared to the original CLIP by simply changing the training recipe, we compare it against other state-of-the-art methods built on CLIP. We compare it to the approaches based on the ResNet-50 architecture, as well as to the approaches based on vision transformers.

Table 2: Comparison of CLIP 🚀 with state-of-the-art zero-shot ImageNet classification accuracy. ResNet-50 backbone, unless otherwise specified. Best results in first block in bold. *: Uses larger YFCC100M subset; †: 50 epochs pre-training.

| Method | Pretraining | | | |
|---|---|---|---|---|
| | CC3M | CC12M | YFCC15M | 29M |
| CLIP (Radford et al., 2021) | 20.6 | 36.5 | 32.7 | 44.2 |
| ProtoCLIP (Chen et al., 2022) | 21.5 | | 32.0 | |
| CyCLIP (Goel et al., 2022) | 22.1 | | | |
| CLOOB (Fürst et al., 2022) | 24.0 | | 35.7 | |
| DeCLIP (Li et al., 2021) | 27.2 | 41.0 | 41.9 | 49.3 |
| CLIP 🚀 (ours) | **27.4** | **44.4** | **43.4** | **53.3** |
| *Using pre-trained models or additional supervision* | | | | |
| BoW (Tejankar et al., 2021) | 30.3 | | | |
| UniCL (Yang et al., 2022) | | | 40.5 | |
| PyramidCLIP (Gao et al., 2022) | | | 43.7 | |
| SoftCLIP (Gao et al., 2023) | 24.2 | 43.2 | | |
| *Using ViT-B/32* | | | | |
| FILIP (Yao et al., 2021) | | | 37.8* | |
| MS-CLIP-S (You et al., 2022) | | | 36.7* | |
| UniCLIP (Lee et al., 2022) | | | 42.8† | 54.2 |

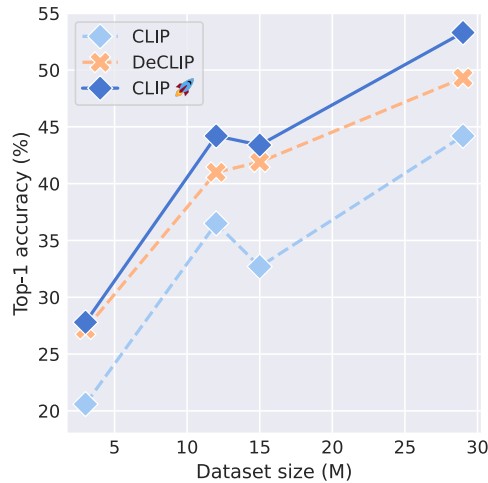

Figure 3: ImageNet performance trend when scaling the dataset size when pre-training on CC3M, CC12M, YFCC15M and Open29M respectively.

Our main comparison is presented in Tab. 2, where we compare the methods based on ResNet-50 or ViT with similar FLOPs *e.g.* ViT-B /32. Analyzing the results in the first block of rows in Tab. 2, CLIP 🚀 not only boosts original CLIP when pre-trained on all datasets, but it also obtains consistently higher accuracy than all the prior works having the same architecture. Broadening the comparison to methods employing pre-trained networks, we observe slightly worse performance, -2.1% and -0.3% on CC3M and YFCC15M. Although negative, this small gap can be considered encouraging as our model could also benefit from initialization and potentially overturn the gap. In a second broader comparison with methods that use ViT-B/32, we find the best results for CLIP 🚀 on YFCC15M (+0.6%), despite the longer pre-training of UniCLIP (50 vs. 32 epochs), and slightly worse results on Open29M (-0.9%). This is explainable as vision transformers have less inductive bias and thus benefit more from a larger dataset size.

In Fig. 3 where we consider the scaling ability of CLIP 🚀 compared to the best competitor DeCLIP (Li et al., 2021) and baseline CLIP, we notice the biggest gap with DeCLIP when pre-training on the largest dataset (+4.0% on Open29M vs. +0.6%, +3.2%, +1.5%, for CC3M, CC12M, and YFCC15M, respectively). This result highlights the scalability of our training recipe, which is a fundamental feature given the prominent research direction toward large-scale training. For this reason, we further solidify this scaling investigation by reporting in Tab. 3 the results when pre-training on PMD46M. Here, we compare CLIP 🚀 with various baselines of CLIP and FLAVA (Singh et al., 2022). For a fair comparison, we use the ViT B/16 vision backbone as in Singh et al. (2022) and train by seeing a comparable number of image-text pairs. The analysis of the results showcases a significant gap between CLIP 🚀, CLIP baseline (-5.8%) and FLAVA (-3.3—5.3%). Moreover, CLIP 🚀 matches the performance of the open-clip implementation trained by Cherti et al. (2023) on LAION (400M image-text pairs) for 13B samples seen. This is a remarkable achievement considering that our model is trained on ∼10x less data with ∼10x less samples seen. Together with the trend shown in Fig. 3 these results validate a consistent boost provided by our recipe when scaling data.

Orthogonally, we also perform scaling on the model size. Results are reported in Tab. 4 and Fig. 4. We make comparisons with prior work using image encoders based on the ViT-{S,B,L}/16 architectures. As it can be observed from the ImageNet zero-shot accuracy (first column), our improved recipe is again very beneficial compared the original recipe for CLIP: boosting accuracy by 8.8%, 7.4%, and 9.5% when using ViT-{S,B,L} backbones, respectively. Moreover, it outperforms SLIP (Mu et al., 2022), by 3.2%, 2.2%, and 2.4% on ViT-{S,B,L} respectively, when pre-trained for the same amount of epochs (25), and when pre-trained for a significantly less epochs, *i.e.*, 32 vs 100, improving by 2.0%, 3.6%, 2.1% on ViT-{S,B,L} respectively. These results further highlight the benefit of our improved training recipe that includes multiple views with separate projectors, which allow the model to be both better performing and more efficient in the number of epochs.

Table 3: Linear evaluation on ImageNet after pre-training on the PMD dataset (unless otherwise specified). Scores of CLIP and FLAVA are taken from Singh et al. (2022), open-clip (Cherti et al., 2023) and the original CLIP paper (Radford et al., 2021). Note that, * indicates training on PMD augmented with ImageNet, CCNews and Book-Corpus, while † marks initialization with pre-trained text and vision encoders. "I-T" stands for image-text pairs.

| Method | Dataset | Batch | I-T seen | Lin eval |
|--------|---------|-------|----------|----------|
| CLIP | 70M | 8K | 1.3B | 73.0 |
| FLAVA | >70M | 8K | 1.3B | 73.5* |
|  | 70M | 8K | >1.3B | 74.3† |
|  | >70M | 8K | >1.3B | 75.5*† |
| CLIP 🚀 | 46M | 4K | 1.5B | **78.8** |
| CLIP | 400M (CLIP-WIT) | 32K | 13B | 80.2 |
| CLIP | 400M (LAION) | 86-88K | 13B | 78.7 |

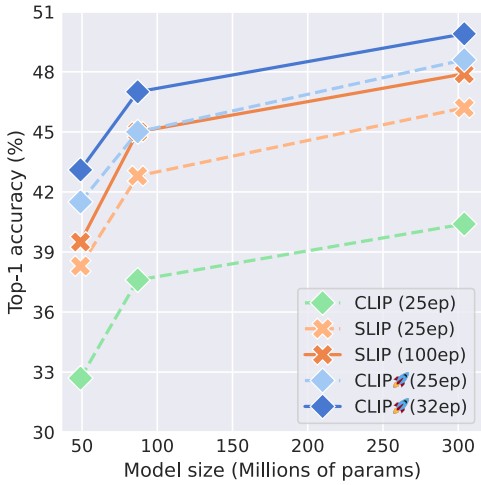

Figure 4: ImageNet zero-shot with ViT-{S,B,L}/16 scaling, trained on YFCC15M.

Table 4: Zero-shot on 23 vision datasets after YFCC15M pre-training for 25 epochs, unless noted otherwise. CLIP and SLIP scores are from the SLIP paper. The average considers also ImageNet, as in Mu et al. (2022).

| | Method | ImageNet | MNIST | Food101 | CIFAR-10 | CIFAR-100 | CUB200-2011 | Stanford Cars | Oxford Aircraft | DTD | Oxford Pets | Caltech-101 | Oxford Flowers | STL10 | EuroSat | RESISC45 | GTSRB | Kitti distance | Country211 | Patch Camelyon | UCF101 | CLEVR counts | Hateful Memes | Rendered SST2 | Average |
|---|---|---|---|---|---|---|---|---|---|---|---|---|---|---|---|---|---|---|---|---|---|---|---|---|---|
| **ViT-S/16** | CLIP | 32.7 | 9.8 | 43.4 | 61.0 | 29.9 | 31.1 | 3.1 | 4.7 | 17.9 | 25.0 | 53.3 | 47.8 | 86.8 | 22.3 | 16.1 | 9.5 | 34.1 | 8.7 | **64.8** | 26.0 | 14.7 | **56.1** | 49.5 | 32.5 |
| | SLIP | 38.3 | **9.9** | 51.6 | **73.0** | **35.4** | 36.3 | 4.2 | 6.1 | 25.7 | **30.9** | 62.8 | 54.3 | 91.6 | **22.4** | 21.9 | **11.0** | 39.9 | 9.6 | 50.8 | 32.8 | 14.8 | 49.6 | 50.1 | 35.8 |
| | CLIP 🚀 | **41.5** | 7.2 | **53.8** | 61.3 | 30.5 | **37.6** | 5.8 | 8.7 | 26.1 | 29.5 | **68.8** | **57.6** | **94.2** | 14.2 | **28.3** | 6.5 | **41.4** | **13.0** | 51.7 | **37.4** | 13.1 | 51.5 | 50.4 | **36.1** |
| | SLIP (100ep) | 39.5 | 2.7 | 53.0 | 68.4 | 39.3 | 36.5 | 4.6 | 5.1 | 26.6 | 33.6 | **68.3** | 55.8 | 91.9 | 18.2 | 22.2 | 13.8 | 38.4 | 8.5 | **62.8** | 33.3 | **19.2** | 51.4 | 49.4 | 36.6 |
| | CLIP 🚀 (32ep) | **43.1** | 9.4 | **57.6** | 62.9 | 34.0 | **40.3** | 5.2 | **10.5** | **28.4** | **34.4** | 68.2 | **59.7** | **93.3** | **19.9** | **22.7** | 8.3 | 36.6 | **11.8** | 50.8 | **37.2** | 14.1 | **53.2** | **51.9** | **37.1** |
| **ViT-B/16** | CLIP | 37.6 | 8.4 | 50.6 | 66.0 | 34.5 | 38.8 | 4.0 | 5.4 | 21.2 | 28.5 | 60.9 | 53.3 | 90.5 | **30.2** | 21.5 | 6.1 | 35.1 | 10.5 | 53.5 | 28.5 | 10.8 | 52.4 | 50.7 | 33.5 |
| | SLIP | 42.8 | **9.8** | 59.5 | **78.6** | 45.2 | 38.7 | 5.4 | 5.7 | 26.1 | **31.1** | 71.0 | 56.6 | 94.4 | 20.3 | 28.9 | **14.5** | 34.0 | 11.6 | **55.4** | 37.7 | 17.5 | 52.8 | **51.1** | 38.6 |
| | CLIP 🚀 | **45.0** | 9.4 | **61.5** | 71.3 | 38.9 | **42.7** | 6.7 | 9.9 | 28.0 | 30.7 | **72.8** | 58.4 | **95.5** | 12.6 | **37.2** | 10.0 | **44.2** | 14.6 | 51.1 | **41.1** | 12.4 | **52.9** | 50.4 | **39.0** |
| | SLIP (100ep) | 45.0 | **17.1** | **63.3** | **79.2** | **50.4** | 44.7 | 8.1 | 8.4 | 26.2 | 34.7 | 74.0 | 61.3 | 95.4 | 20.8 | 27.8 | 11.7 | 35.2 | 11.5 | **52.1** | 37.1 | 13.0 | **55.1** | 49.9 | 39.0 |
| | CLIP 🚀 (32ep) | **47.0** | 9.8 | 63.0 | 67.1 | 37.8 | **45.7** | **8.2** | **11.2** | **30.3** | **35.4** | **75.4** | **62.8** | 95.1 | **23.2** | **36.0** | 8.1 | **35.6** | **15.6** | 51.6 | **42.7** | **13.5** | 54.3 | **50.1** | **40.0** |
| **ViT-L/16** | CLIP | 40.4 | 10.3 | 59.5 | 72.9 | 41.5 | 40.3 | 6.9 | 6.4 | 20.6 | 27.9 | 65.4 | 55 | 94.2 | 22.7 | 28.8 | 5.8 | **41.4** | 12.6 | **54.9** | 34.3 | 12.9 | **54.3** | 50.1 | 37.4 |
| | SLIP | 46.2 | 13.2 | 64.4 | **87.8** | **56.4** | 39.8 | 8.6 | 7.8 | 26.8 | 32 | 76.6 | 59.4 | 96.6 | **27.7** | 36.5 | 7.2 | 28.8 | 15.6 | 54.4 | 42.6 | 14.1 | 53.4 | 50.1 | 41.1 |
| | CLIP 🚀 | **48.6** | **16.3** | 69.1 | 83.5 | 49.9 | **46.2** | 9.0 | 11.8 | 30.4 | 32.9 | 78.5 | 63.6 | **96.7** | 20.7 | 41.3 | 8.4 | 36.0 | **18.2** | 51.5 | **43.2** | 13.8 | 52.0 | 50.1 | **42.2** |
| | SLIP (100ep) | 47.9 | **14.9** | 69.2 | **87.5** | 54.2 | 39.8 | 9.0 | 9.5 | 29.9 | 41.6 | **80.9** | 60.2 | 96.2 | **34.5** | **46.0** | 8.6 | **30.7** | 14.2 | 50.6 | **44.1** | **17.4** | **55.0** | 49.8 | 43.1 |
| | CLIP 🚀 (32ep) | **50.0** | 12.5 | **70.3** | 84.7 | 52.0 | **46.9** | **11.8** | **11.2** | **32.1** | **41.7** | 77.5 | **68.3** | **97.0** | 28.7 | 44.4 | **10.2** | 27.9 | 17.9 | **56.2** | 44.0 | 13.8 | 52.0 | **50.2** | **43.5** |

Next, we consider the generalization of CLIP 🚀 to different types of vision benchmarks. To do so, we follow the setting proposed by Radford et al. (2021), testing the zero-shot transfer of CLIP 🚀 representations on 22 vision benchmarks (plus ImageNet) as reported in Tab. 4. We observe an improvement of the average performance (see last column), in all the settings considered, *i.e.*, ViT-{S,B,L} with 25 or more pre-training epochs. In particular, focusing on the 25 epochs pre-training, CLIP 🚀 provides a boost in most of the benchmarks; it improves over baseline CLIP in 18/23, 20/23, and 18/23 benchmarks with ViT-{S,B,L}, respectively. Moreover, compared with SLIP, regardless of the pre-training duration, CLIP 🚀 obtains the best performance in 13 to 16 of the 23 benchmarks. As a final note, we notice that the types of benchmarks positively affected by our recipe remain almost the same across different ViT sizes and number of epochs. This might be explained by the induced invariances of the strong data augmentations used in our recipe.

### 5.3.3 Ablations

We quantify the impact of the different components of our improved training recipe by isolating and ablating them. We first ablate the regularization techniques, and then the projector design. Finally, we analyze the computational cost of our recipe. For all the ablations, we use CLIP 🚀 pre-trained on YFCC15M with ResNet-50 as the vision backbone. For these experiments, we use the zero-shot ImageNet top-1 classification accuracy as a metric. Also, note that not all the ablations were run on the final recipe: some of them have been performed earlier in our exploration, and thus provide results different from the one reported in Sec. 5.3.2.

(a) Text augmentation

| EDA | RMSW | Multi-cap | Top-1 |
|---|---|---|---|
|  |  |  | 42.8 |
| ✓ |  |  | 43.0 |
| ✓ | ✓ |  | 43.1 |
| ✓ | ✓ | ✓ | **43.4** |

(b) Vision augmentation

| SimCLR aug. | Multi-view | BYOL aug. | Top-1 |
|---|---|---|---|
|  |  |  | 32.7 |
| ✓ |  |  | 35.2 |
| ✓ | ✓ |  | **43.4** |
| ✓ | ✓ | ✓ | 43.2 |

(c) Projectors design

| Text-proj | Vision-proj | Top-1 |
|---|---|---|
| $\text{Lin}^{\text{T-shared}}$ | $\text{Lin}^{\text{V-shared}}$ | 40.9 |
| $\text{MLP}^{\text{T-shared}}$ | $\text{MLP}^{\text{V-shared}}$ | 39.6 |
| $\text{Lin}^{\text{T}}$ | $\text{Lin}^{\text{V}}$ | 41.0 |
| $\text{MLP}^{\text{T}}$ | $\text{Lin}^{\text{V}}$ | 41.0 |
| $\text{Lin}^{\text{T}}$ | $\text{MLP}^{\text{V}}$ | 42.6 |
| $\text{MLP}^{\text{T}}$ | $\text{MLP}^{\text{V}}$ | **43.1** |

(d) Label smoothing

(e) Dropout

(f) Efficiency

Figure 5: Ablations. (b) EDA: Easy Data Augmentation (Wei & Zou, 2019); RMSW: remove stop-words (Tejankar et al., 2021); Multi-cap: random multi-caption sampling; (e) the accuracy is relative to having dropout set to 0; (f) the reported values were computed using eight Nvidia V100-SMX2 32GB GPUs and our recipe with ResNet-50 backbone (see Sec. 5.2). The annotation of each point indicates (# of views, view resolution).

**Regularization.** In Fig. 5a and 5b we break down the data augmentation recipe into its main components. For the text augmentation recipe (see Fig. 5a), we evaluate the impact of EDA (Wei & Zou, 2019), removing stop-words (RMSW) (Tejankar et al., 2021), and random multi-caption sampling (see Sec. 5.2). We notice that each of the three techniques provides a small boost, summing up in total to +0.6% to CLIP 🚀. In particular, while RMSW only provides +0.1% it is helpful to reduce noisy/uninformative image-text captions from the training. In general, we may explain the relatively small boost given by text augmentations as our recipe is more focused on the vision part due to the introduction of multiple views. Indeed, in Fig. 5b we showcase the improvement of using multiple views combined with SimCLR-style (Chen et al., 2020) image transformations that results in +10.7% *w.r.t.* the original CLIP. On the contrary, the use of stronger and asymmetric BYOL-like (Grill et al., 2020) image transformations are not beneficial in the multi-modal CLIP pre-training. We conjecture that the implicit regularization provided by these augmentations is too strong in the presence of the noisy supervisory signal provided by image-text pairs.

In addition to implicit data augmentation regularization, we also study the effect of the explicit regularization techniques present in our improved recipe in Fig. 5d and 5e. Specifically, Fig. 5d depicts the trend of the accuracy (blue line) when increasing label smoothing in the contrastive computation between strong augmentations. From the plot, it is easy to observe that higher smoothing factors generate better results, +2.8% by increasing the smoothing from 0.0 to 0.1. To explain this significant boost, we report in the same plot the trend of the logit scale (orange curve), which corresponds to the inverse of the learned temperature, $1/\tau$. The logit scale is negatively correlated with the label smoothing factor, suggesting that label smoothing improves accuracy by mitigating overfitting as indicated by reduced logit scales. This is also corroborated by concurrent work (Gao et al., 2023). In Fig. 5e, we analyze the effect of the dropout probability in the text encoder when pre-training on three different datasets. We find that on YFCC15M, the larger and more noisy dataset among the three, applying dropout is detrimental. On the contrary, as the datasets become smaller and less noisy, dropout provides slight accuracy boosts.

**Projector design.** A crucial modification that we propose in our improved recipe is the use of two separated non-linear projectors to project strong/local augmentations/views into the CLIP cross-modal embedding. We ablate the possible architectural choices for the projector in Fig. 5c. In the first two rows, the same projectors are used for the weakly and strongly augmented text, and similarly for the image projectors. We observe that with such shared projectors having a linear projection (`Lin`) is beneficial compared to non-linear (`MLP`). However, in the following rows, we show that if we employ separate projectors for the weakly- and the strongly-augmented text and images, using non-linear projections (last row, `MLP`$^T$ and `MLP`$^V$) produces a boost of $+2.2\%$ from the baseline case (first row, shared linear projection). This finding is perhaps not surprising as strongly augmented text-image pairs are more challenging to align, providing enough regularization such that the MLPs can be trained without overfitting.

**Efficiency.** Our improved training recipe increases the computational costs due to the simultaneous use of multiple image and text views/augmentations, combined with more complex projectors. In Fig. 5f, we show how the accuracy and the training time are impacted by the resolution of the multiple image views. As can be noted, we can reduce the training time by using lower resolution image views. However, we observe that decreasing the resolution of the image views impacts negatively the performance. In particular, substantial downsampling *i.e.*, $< 128$, causes a drastic drop in the performance, while less aggressive downsampling, *i.e.*, $192^2$ and $160^2$, still provides results competitive with the state of the art, but at a training cost reduced by $10\%$ to $20\%$.

## 6 Related work

**Contrastive vision-language pre-training.** Seminal works like CLIP (Radford et al., 2021) and ALIGN (Jia et al., 2021) have proven how contrastive VLP at scale provides rich and general representations for a multitude of downstream tasks. However, these methods apply the contrastive loss only between the entire image and its caption, ignoring possible local alignments, and do not use distorted views of the input data limiting the robustness of the learned representations. To this end, FILIP (Yao et al., 2021) and PyramidCLIP (Gao et al., 2022), have tried to refine the contrastive loss to consider image and text parts (patch-tokens/tags pairs), and SoftCLIP (Gao et al., 2023) use image-text parts alignment combined with a softening loss computed over the negatives pairs. Instead, other works tackled a better robustness of the learned representations: CLOOB (Fürst et al., 2022) using Hopfield networks to regulate the covariance of the learned representations, MS-CLIP (You et al., 2022) and FIBER (Dou et al., 2022), tried to overcome the use of dual modality encoders in CLIP to obtain better fused multimodal features, while FLIP (Li et al., 2022b) and MaskCLIP (Dong et al., 2022) used image masking. More related to CLIP 🚀, some works improved representation robustness using multiple views: SLIP (Mu et al., 2022) used multiple views of images, but unlike us, they were used to optimize an auxiliary unimodal SSL contrastive loss; UniCLIP (Lee et al., 2022) and ViewCo (Ren et al., 2023) instead, used multiple image views for both the unimodal and multimodal contrastive losses. However, this is different from CLIP 🚀 since we do not use unimodal losses while we also use multiple views of the text to generate aligned weakly and strongly augmented image-text pairs. Finally, since image-text pairs are often noisy in web-crawled datasets, UniCL (Yang et al., 2022) proposed a solution that integrates image-label pairs from supervised datasets, while CoCa (Yu et al., 2022), OTTER (Wu et al., 2021), BLIP (Li et al., 2022a), and BoW-CLIP (Tejankar et al., 2021) proposed to denoise text captions by generating them or substituting them with bag of words (BoW). Inspired by BoW-CLIP, in CLIP 🚀, we also remove stop-words from the image captions.

**SSL-inspired non-contrastive losses in VLP.** All the aforementioned contrastive VLP approaches are highly sensitive to the quality and diversity of negative samples present in the batch. To overcome this issue, several recent works have explored the use non-contrastive losses inspired by recent SSL advancements. However, as shown by the non-contrastive baselines in our experiments, none of the methods in the literature obtained better results without keeping the standard CLIP contrastive loss. Differently to our baselines, some existing methods only adopt non-contrastive losses for the unimodal encoders: DeCLIP (Li et al., 2021) use a consistency loss for the image encoder, while VoLTA (Pramanick et al., 2022) adopt the Barlow Twins loss (Zbontar et al., 2021) separately on both the vision and text encoders. More similar to our baselines some works integrates the CLIP contrastive loss with a multimodal non-contrastive loss: ProtoCLIP (Chen

et al., 2022) and xCLIP (Zhou et al., 2022) adopt a clustering-based loss, while CyCLIP (Goel et al., 2022) combines both unimodal and multimodal consistency losses for vision and text. However, these methods employ custom non-constrastive losses that do not reflect any existing SSL approaches such as our baselines. Finally, FLAVA (Singh et al., 2022) and ViCHA (Shukor et al., 2022) use image and text masking combined with multimodal masking and/or image-text matching, but the masking approach is out of the scope of this work as it usually requires fine-tuning, as shown by He et al. (2022), while we focus on zero-shot transfer.

## 7 Conclusion

In conclusion, this paper investigated the effectiveness of combining contrastive learning with recent advances in self-supervised and supervised learning. Our study includes several baselines using successful non-contrastive loss functions from visual self-supervised learning, and compares their performance with CLIP, a seminal work in this area. Our findings indicate that the SSL-inspired baselines outperform a basic implementation of CLIP, but the advantage disappears when an improved training recipe is employed. Noticeably, we show that a simple CLIP baseline can be substantially improved by applying this improved training recipe, composed of image and text augmentations, label smoothing, dropout, and deeper projector networks. Our improved training recipe for CLIP achieves state-of-the-art performance on four datasets and outperforms related methods by significant margins.

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

## A  Modified SwALIP

To mitigate the instability issues of SwALIP (see Sec. 4), we develop a slightly modified version that does not exhibit learned prototypes, rather, it uses representations from the other modality as prototypes, and clusters samples in the current modality around those. In practice, in order to implement this, we need multiple views of each image-text pair as in our improved recipe, where we have two strongly-augmented views. Considering a batch of samples containing such two sets of correlated views projected into latent space, $\mathcal{Z} = \{(\boldsymbol{z}_i^A, \boldsymbol{z}_i^B)\}_{i=1}^N$ and $\bar{\mathcal{Z}} = \{(\bar{\boldsymbol{z}}_i^A, \bar{\boldsymbol{z}}_i^B)\}_{i=1}^N$, the new formulation reads as follows:

$$\mathcal{L}_{\text{NC}}^{A \to B} = \mathcal{L}_{\text{XE}}^{A \to B}(\mathcal{Z}^A, \bar{\mathcal{A}}^A, \mathcal{P}^B), \quad \bar{\mathcal{A}}^A = \lambda \operatorname{sk}(\operatorname{sim}(\bar{\mathcal{Z}}^A, \mathcal{P}^B)) + (1 - \lambda)\mathcal{I}, \quad \mathcal{P}^B = \mathcal{Z}^B \tag{7}$$

$$\mathcal{L}_{\text{NC}}^{A \to B} = -\sum_{i=1}^N \sum_{k=1}^N \bar{\boldsymbol{a}}_{i,k}^A \log \frac{\exp\left(\operatorname{sim}\left(\boldsymbol{z}_i^A, \boldsymbol{p}_k^B\right)/\tau\right)}{\sum_{k'=1}^N \exp\left(\operatorname{sim}\left(\boldsymbol{z}_i^A, \boldsymbol{p}_{k'}^B\right)/\tau\right)}, \tag{8}$$

where $\bar{\boldsymbol{a}}_i^A \in \bar{\mathcal{A}}^A$ is the vector of cluster assignments for the correlated view, $\bar{\boldsymbol{z}}_i^A$, of $\boldsymbol{z}_i^A$ in one modality, *e.g.*, image, and each prototype $\boldsymbol{p}_k^B \in \mathcal{P}^B$ corresponds to one view of the samples, $\boldsymbol{z}_k^B$, in the other modality, *e.g.*, text. Assignment vectors are obtained by applying Sinkhorn-Knopp (sk) on the multimodal similarity matrix computed against the prototypes $\mathcal{P}^B$. The loss is first computed using the swapped assignments typical of SwAV, *i.e.*, $\mathcal{L}_{\text{XE}}^{A \to B}(\mathcal{Z}^A, \bar{\mathcal{A}}^A, \mathcal{P}^B)$ and $\mathcal{L}_{\text{XE}}^{A \to B}(\bar{\mathcal{Z}}^A, \mathcal{A}^A, \mathcal{P}^B)$, and then in both directions $A \to B$ and $B \to A$, so that it becomes symmetric. Also, since the view of the samples in the "other" modality used

as prototypes is arbitrary, *i.e.*, $\mathcal{P}^B = \mathcal{Z}^B$ and $\bar{\mathcal{P}}^B = \bar{\mathcal{Z}}^B$ are both valid, in our implementation we average the loss considering both cases. Finally, to stabilize training and avoid collapsed solutions we also average the output of the Sinkhorn-Knopp algorithm with the identity matrix, $\mathcal{I}$, to strengthen relationships on the diagonal (ground truth image-text pairs).

## B   Additional results

Table 5: Zero-shot transfer on ImageNet and several vision datasets taken from Radford et al. (2021). Pre-training on YFCC15M for 32 epochs with ResNet-50 as vision backbone.

| Method | ImageNet | MNIST | Food101 | CIFAR-10 | CIFAR-100 | CUB200-2011 | Stanford Cars | Oxford Aircraft | DTD | Oxford Pets | Caltech-101 | Oxford Flowers | STL10 | EuroSat | RESISC45 | GTSRB | Kitti distance | Country211 | Patch Camelyon | UCF101 | CLEVR counts | Hateful Memes | Rendered SST2 | Average |
|---|---|---|---|---|---|---|---|---|---|---|---|---|---|---|---|---|---|---|---|---|---|---|---|---|
| CLIP 🚀 | **43.1** | 9.8 | 53.3 | 47.0 | 21.5 | 37.8 | 6.2 | 8.9 | 25.7 | 34.1 | 66.6 | **59.6** | 89.7 | 18.3 | 24.0 | 8.3 | 37.0 | **11.8** | 51.0 | **37.6** | 12.9 | 52.7 | 50.1 | 35.1 |
| SwaLIP 🚀 | 42.5 | **10.8** | 52.6 | 52.5 | 21.8 | 37.8 | 6.5 | 9.6 | 27.9 | **34.6** | 67.1 | 58.3 | 89.0 | 14.6 | 19.4 | **11.3** | **41.1** | 11.0 | 50.2 | 37.0 | **13.3** | 52.9 | 50.1 | 35.3 |
| BarLIP 🚀 | 42.6 | 10.7 | **53.5** | 51.0 | 19.4 | 38.3 | 6.1 | **10.4** | 28.9 | 32.2 | 65.7 | 56.1 | 90.7 | 13.2 | **26.9** | 8.4 | 39.8 | 11.4 | 50.9 | 33.7 | 12.8 | **57.0** | **50.7** | 35.2 |
| SiamLIP 🚀 | 42.8 | 10.7 | 52.7 | 46.4 | **22.3** | **39.1** | 5.1 | 7.6 | **29.8** | 33.5 | 66.4 | 58.6 | **91.9** | 12.5 | 25.9 | 9.1 | 39.2 | 11.6 | 51.4 | 36.0 | 12.4 | 53.9 | 49.9 | 35.2 |
| BYOLIP 🚀 | 42.8 | 10.6 | 50.8 | **54.1** | **22.3** | **39.1** | **6.9** | 7.9 | 27.2 | 32.4 | **68.2** | 59.3 | 90.8 | **19.7** | 23.1 | 7.8 | 38.6 | 11.2 | **52.5** | 36.6 | 12.9 | 53.2 | 50.1 | **35.6** |

**Generalization of non-contrastive baselines**   In the main paper we compared our improved baselines (see Sec. 4) with CLIP 🚀 in terms of zero-shot image classification accuracy on ImageNet. However, one might ask whether the use of those non-contrastive losses is beneficial in terms of generalization. In Tab. 5, we provide an answer to this question, expanding the evaluation to a larger set of 22 vision datasets. We observe that the best results on the downstream datasets are rather spread among the different improved baselines. It is hard to find a model overcoming others consistently, despite BYOLIP presents a slightly better overall average. In general, this result confirm our finding that it is not worth to complicate the model by using multiple objectives, but it is enough to use our well-tuned recipe, CLIP 🚀.

**Sensitivity analysis wrt. $\alpha$ and $\beta$**   Here, we report an ablation of the $\alpha$ and $\beta$ hyper-parameters regulating the weights of the contrastive and non-contrastive terms, respectively (see Eq. 6). To reduce the computational burden, we carry out the analysis only for BarLIP 🚀, pre-trained with ResNet-50 on YFCC15M for 32 epochs. We ablate values of $\beta$ while keeping $\alpha$ fixed at 1 and vice versa. Results are reported in Table 6, where we observe that increasing the value of $\beta$ with $\alpha$ fixed at 1 slightly harms the performance of up to 0.8%, while doing the opposite, i.e., increasing $\alpha$ with $\beta$ fixed at 1, leads to a substantial drop of accuracy, up to 3.7%. Notably, using only the non-contrastive loss ($\alpha = 0, \beta = 1$) is problematic for zero-shot evaluation because the CLIP projector is not optimized and it is hard to perform zero-shot classification on the non-contrastive projector. These results further confirm the ineffectiveness of adopting non-contrastive losses together with our improved recipe.

Table 6: Sensitivity to loss weights. ImageNet zero-shot results obtained with BarLIP 🚀.

| $\alpha$ | $\beta$ | YFCC15M |
|---|---|---|
| 1 | 0 | 43.2 |
| 1 | 0.25 | 42.8 |
| 1 | 0.50 | 42.9 |
| 1 | 0.75 | 42.4 |
| 0 | 1 | 0.1 |
| 0.25 | 1 | 38.9 |
| 0.50 | 1 | 41.6 |
| 0.75 | 1 | 42.5 |
| 1 | 1 | 42.6 |

## C   Pseudo-code for multi-view

In Pseudo-Code 3, we show the actual implementation adopted in CLIP 🚀, when multiple (`num_augs`) strong augmentations are adopted. It is important to note that when we compute the contrastive among strongly-augmented image-text pairs, each strong augmentation is contrasted with all the strong augmentations of the other modality. For instance, given two strong augmentations (`num_augs = 2`), each image augmentation is contrasted with both the strong text augmentations, and vice versa.

**Pseudo-code 3** Detailed implementation of CLIP 🚀.

```python
# a, b: image and its corresponding text
# f_t, f_v: vision and text encoders with (in_dim, embed_dim)
# lin, mlp: Linear and MLP(hidden_dim) with (embed_dim, proj_dim)
# tau_w, tau_s: learnable temperature for weak and strong augmentations
# sim_target: identity matrix

proj_v_w, proj_t_w = lin(f_v(x), bias=None), lin(f_t(x), bias=None)
proj_v_s, proj_t_s = normalize(mlp(f_v(x))), normalize(mlp(f_t(x)))
def training_step(a, b):

    # weak augmentations
    a_w, b_w = aug_v(a, 'weak'), aug_t(b, 'weak')

    # strong augmentations
    a_s, b_s = [], []
    for _ to range(num_augs):
        a_s += aug_v(a, 'strong')
        b_s += aug_t(b, 'strong')

    # projecting into a shared space
    z_a_w, z_b_w = proj_v_w(a_w), proj_t_w(b_w)
    z_a_s, z_b_s = proj_v_s(*a_s), proj_t_s(*b_s)

    # CLIP loss weak augmentation
    sim_a_to_b_w = tau_w * z_a_w @ z_b_w.t()
    sim_b_to_a_w = tau_w * z_b_w @ z_a_w.t()
    loss_a_w = cross_entropy(sim_a_to_b_w, sim_target)
    loss_b_w = cross_entropy(sim_b_to_a_w, sim_target)

    # CLIP loss strong augmentation
    sim_a_to_b_s = []
    sim_b_to_a_s = []
    for i in range(num_augs):
        sim_a_to_b_s += [tau_s * x @ z_b_s[i].t() for x in z_a_s]
        sim_b_to_a_s += [tau_s * x @ z_a_s[i].t() for x in z_b_s]
    loss_a_s = sum([cross_entropy(x, sim_target, label_smoothing) for x in sim_a_to_b_s]) / len(sim_a_to_b_s)
    loss_b_s = sum([cross_entropy(x, sim_target, label_smoothing) for x in sim_b_to_a_s]) / len(sim_b_to_a_s)

    loss_a = (loss_a_w + loss_a_s * num_augs) / (1 + num_augs)
    loss_b = (loss_b_w + loss_b_s * num_augs) / (1 + num_augs)
    loss = (loss_a + loss_b) / 2

    return loss
```

