# OpenReview forum: "Improved baselines for vision-language pre-training"
_TMLR — Accepted by TMLR_

### Review · Reviewer_sXof · 2023-05-30

**Summary Of Contributions:**

The authors critically analyze improved self-supervised learning procedures that claim to improve over CLIP contrastive learning. They show that many of these methods do not improve over the vanilla CLIP model when better data augmentations (and other improvements) are applied. The result is a simpler training objective that outperforms more complicated methods. They conduct extensive analysis to justify their design choices, such as augmentation strategies and projection models.

**Audience:**

Yes

**Broader Impact Concerns:**

I have no major concerns about the ethical implications of the work: it is research on self-supervised learning, primarily applied downstream to image classification. It is an analysis paper which does not introduce significant risk.

**Claims And Evidence:**

Yes

**Requested Changes:**

**Strengthen the work:** As mentioned above, I think that including further scaling experiments (along the model parameters dimension) would be very valuable to the paper.

I think the paper in general is very polished and well written, and do not have any other requested changes.

**Strengths And Weaknesses:**

## Strengths:
- Proposes a simple yet effective improved training recipe which involves improved image and text augmentations, label smoothing, and stronger projector models. The strong and weak losses are averaged.
- Demonstrates the effectiveness of the training strategy over most complicated training objectives. The proposed method outperforms CLIP models optimized on more complex loss functions (such as the consistency-based objectives, the Barlow Twins loss, or clustering criteria).
- The proposed method has a convincing scaling curve (Figure 3), which suggests that the gains will still hold when models are trained on even more data (i.e., the proposed method will still be helpful even when the full CLIP 400M dataset is available).
- The paper conducts careful ablations on the objectives and projections to justify design choices.


## Weaknesses:
- There are limited experiments on scaling the CLIP model. Table 3 shows some favorable results with ViT-B against ViT-S. It would be beneficial to conduct even more scaling experiments, to show that the proposed method does indeed benefit models even at larger parameter sizes. If possible, having ViT-L, ResNet-101 and ResNet-152 results would make the findings substantially stronger. It would also be valuable to scale the text encoder similarly.

---

> ### Author Response · Authors · 2023-07-09
> **Response to Reviewer sXof**
>
> We thank the Reviewer for the feedback and we highly appreciate the positive reception of our work.
> As suggested, we ran additional experiments to demonstrate the effectiveness of our improved recipe when scaling the model size. We pre-trained CLIP🚀 on YFCC15M using ResNet-101 and ViT-L/16 as vision backbone, and report the ImageNet-1k zero-shot accuracy, while keeping fixed all hyper-parameters. The obtained results are reported in Table B, and we will include these experiments in the paper. Moreover, given the multiple data points available for ViT we plot them to visualize trends, see Figure A.
>
> Table B. Further scaling of the model size. Reported metrics is zero-shot top-1 Imagenet accuracy.
>
> | Method     | Epochs | YFCC15M  |
> |:---------- |:------:|:--------:|
> | _ResNet-101_ |        |          |
> | CLIP       | 32     | 36.7     |
> | CLIP🚀     | 32     | **45.0** |
> |            |        |          |
> | _ViT-L/16_   |        |          |
> | CLIP       | 25     | 40.4     |
> | SLIP       | 25     | 46.2     |
> |            | 100    | 47.9     |
> | CLIP🚀     | 25     | **48.6** |
> |            | 32     | **49.9** |
>
> [Figure A](https://docs.google.com/document/d/e/2PACX-1vQNM-1kKBZtxetCbl7XtDxMWJ341wDW9-als8LON4s9HkDp2xo1p9K3O9UpeAVqF7YZFsb9VhecWAxd/pub)
>
> From the results with ResNet-101 it can be observed a consistent positive gap compared to baseline CLIP trained with the same vision backbone, +8.3%. Also, we outperform CLIP🚀 with ResNet-50 (43.2%) of +1.8%, using the exact same hyper-parameters. Next, observing Figure A, we notice coherent improvements when scaling ViT /16 size; the trends highlight the superiority of CLIP🚀vs SLIP even when trained for much less epochs (25 or 32 vs 100, respectively).

---

> > ### Author Response · Authors · 2023-08-19
> >
> > Dear Reviewer sXof, please let us know if our response solves the raised concerns.

---

### Review · Reviewer_WDqT · 2023-06-06

**Summary Of Contributions:**

This paper conducts a large number of experiments to analyze the effect of combining CLIP with various self-supervised representation learning techniques. Experiments show that combining different self-supervised representation learning techniques can achieve performance gains on downstream tasks compared to the original implementation of CLIP. In addition, this paper also proposes a advanced training recipe, which has achieved significant improvements compared to the original implementation. However, when combined with self-supervised techniques, there is no obvious improvement.

**Audience:**

Yes

**Broader Impact Concerns:**

Nan

**Claims And Evidence:**

Yes

**Requested Changes:**

1. If the data scale is further expanded, is the advanced recipe better than the original implementation?
2. please present more analysis of experimental results.

**Strengths And Weaknesses:**

Strengths:
1. The training recipe proposed in this paper is very instructive for practice and saves a lot of hyper-parameter search work.
2. Many techniques, including image augmentation, text augmentation, label smooth, projector design, have been empirically proven to improve representation quality.
3. Extensive ablation experiments analyze the effect of each component.

Weaknesses:
1. The significance of this paper is worthy of recognition, but many experimental phenomena have not been analyzed in detail. For instance, why does applying self-supervised representation learning techniques not lead to further performance gains when using advanced recipe? I'm guessing it might have something to do with the use of image and text augmentation.
2. Considering that the title of the transaction includes "Machine Learning Research", a qualified paper should not only have comprehensive and impressive experiments, but also include explanations of experimental phenomena and some theoretical analysis. I believe doing so will greatly improve the quality of the article and make it more meaningful to the community.

---

> ### Author Response · Authors · 2023-07-09
> **Response to Reviewer WDqT - part 1**
>
> We acknowledge the reviewer for the feedback provided. We are pleased that reviewer WDqT recognizes the significance and interest of our work for the community, and we are happy to address the critiques that were raised.
>
> 1. **[Further scaling of data]** We conducted additional experiments using a larger dataset, a 46M subset of PMD [1], containing image-text pairs gathered from different data sources including Open29M and others (see [1] for further details). Note that we use a smaller subset (46M out of 70M) as we were not able to reproduce part of the PMD dataset. On this dataset we compare CLIP 🚀 with various versions of CLIP and FLAVA [1], a vision-language model relying on the multimodal contrastive loss combined with the multi- and uni-modal masking losses and the image-text matching loss. To guarantee a fair comparison we follow the experimental setup of FLAVA [1]; our model is trained with the ViT-B/16 vision backbone and processes/sees a comparable number of image-text pairs as FLAVA throughout the training. The evaluation is carried out with linear probing on ImageNet. We report the top-1 validation accuracy on ImageNet in Table A, where scores of CLIP and FLAVA are taken from [1], open-clip or and the original CLIP paper.
>
> Table A. Linear evaluation on ImageNet after pre-training on PMD [1] dataset (unless otherwise specified). Note that, "*" indicates training on PMD augmented with ImageNet, CCNews and BookCorpus, while "†" marks initialization with pre-trained text and vision encoders.
>
> | Method        | Dataset size    | Batch size | Samples seen | Linear eval acc@1 |
> |:------------- |:--------------- |:----------:|:------------:|:------------:|
> | CLIP [1]      | 70M             | 8K         | 1.3B         | 73.0         |
> | FLAVA [1]     | >70M            | 8K         | 1.3B         | 73.5*        |
> |               | 70M             | 8K         | >1.3B        | 74.3†        |
> |               | >70M            | 8K         | >1.3B        | 75.5*†       |
> | CLIP🚀 (ours) | 46M             | 4K         | 1.5B         | **78.8**     |
> |               |                 |            |              |              |
> | CLIP [2]      | 400M (CLIP-WIT) | 32K        | 13B          | 80.2         |
> | CLIP [3]      | 400M (LAION)    | 86-88K     | 13B          | 78.7         |
>
> The results showcase a remarkable gap between CLIP 🚀, CLIP [1] baseline (-5.8%) and FLAVA [1] (-3.3—5.3%). Moreover, CLIP 🚀 matches the performance of the open-clip implementation trained on LAION (400M image-text pairs) for 13B samples seen. This is a great achievement, considering that our model is trained on ~10x less data with ~10x less samples seen. Combined with the scaling trend presented in Figure 3 of the main paper, these results highlight the consistent boost provided by our recipe when scaling the data size. Intuitively, we might explain such a consistent boost as adding new data widens the support of the data distribution, and the augmentations present in our recipe help making the support of data distribution around the training samples more dense.
>
> [1] Singh, Amanpreet, et al. "Flava: A foundational language and vision alignment model." _Proceedings of the IEEE/CVF Conference on Computer Vision and Pattern Recognition_. 2022.
>
> [2] Radford, Alec, et al. "Learning transferable visual models from natural language supervision." _International conference on machine learning_. PMLR, 2021.
>
> [3] Cherti, Mehdi, et al. "Reproducible scaling laws for contrastive language-image learning." _Proceedings of the IEEE/CVF Conference on Computer Vision and Pattern Recognition_. 2023.

---

> > ### Author Response · Authors · 2023-07-09
> > **Response to Reviewer WDqT - part 2**
> >
> > 2. **[Lack of theoretical analysis/explanations]** We understand the point of the reviewer, but we would remark that, despite the title of the journal, the [TMLR call for papers]((https://jmlr.org/tmlr/editorial-policies.html)) reads:
> >
> > > new algorithms with sound empirical validation, **optionally** with justification of theoretical, psychological, or biological nature.
> >
> > Our paper focuses on resetting the state of the art after the advent of many SSL-inspired solutions that claimed superiority w.r.t. CLIP.
> >
> > We have hypotheses as to why self-supervised losses may underperform in our setting:
> > * CLIP is mostly used in the “zero-shot” mode at inference time. This means that given an image, the model needs to choose the most similar caption/label among many different ones. Note that this is very similar to how the contrastive loss is constructed, where a batch of images is compared to a batch of texts, and for each sample we supervise using the ground truth pair in the other modality. Yet, the non-contrastive objectives that we tested are constructed differently:
> >     * BYOLIP and SiamLIP only attract positives but have no interaction between samples in the batch, so they have no way to guarantee that the correct pair is the closest
> >     * Differently, BarLIP provides a way for the samples in the batch to interact through cross-correlation. However, this requires the features to be standardized instead of l2-normalized, which makes it hard to find a suitable distance function to use at inference time
> >     * SwALIP, instead, groups samples into clusters, pulling them close to the cluster prototype / centroid. While this might be good for learning representations, it is detrimental for zero shot evaluation, which instead requires separation of samples in the feature space.
> >
> >     We believe that zero-shot evaluation benefits particularly the contrastive loss-based approaches.
> >
> > * We also hypothesize that the additional non-contrastive losses work as regularizations. This is supported by two observations made when training without improved training recipe:
> >     * They work better on networks with less inductive bias (ViT vs ResNet) which often need more regularization
> >     * The advantage tends to disappear when scaling up the when scaling up the dataset size, i.e. when the network “overfits” less, and therefore regularization is less essential
> >
> >
> > As mentioned, these are hypotheses, and proving them is challenging. For each of these points a new paper could be written. We believe this goes beyond the scope of the empirical investigation we provide in our paper.

---

### Review · Reviewer_Z9sJ · 2023-06-28

**Summary Of Contributions:**

Building on the success of contrastive vision language models (primarily CLIP) and self-supervised pretraining strategies, the paper exercises the option of combining the two for better vision language training. For this purpose, the authors propose to extend normal CLIP training with the addition of four popular self-supervised learning (SSL) non-contrastive losses. The extension showed improved vision-language representation learning and this is manifested in the improved model’s capability of zero-shot transfer to benchmark datasets. The authors, though, exercised additional option of sticking to contrastive loss only (i.e., normal CLIP style training) but using stronger training recipes well-known in other works but not employed in CLIP. The second framework yielded even better performance at the cost of slightly increased computation. The experimental design was well thought and the experiments were well-executed. The relevant literature was well studied and summarized with pinpointed differences and similarities of the proposed work with them. In summary, with the simplicity in the training pipeline, the presentation and the easy-flowing experiments, I think the work can make an useful contribution to the community.

**Audience:**

Yes

**Broader Impact Concerns:**

I can’t think of any immediate concerns. However, it is good to have a discussion on it.

**Claims And Evidence:**

Yes

**Requested Changes:**

1. One sensitivity analysis related to weaknesses (point 1) can be added.
2. A few typos need to be corrected.
 - Page 2, second line: The acronym SSL is not defined earlier. Though, it can be interpreted, still it is better to define the term. I have seen Semi-supervised as well as Self-supervised learning both getting the same acronym SSL.
 - Page 5, last line: ‘when computing the contrastive between’ -> ‘when computing the contrastive loss between’

**Strengths And Weaknesses:**

Strengths:
1. This work is a welcome deviation from traditional leaderboard chase. It takes a step back and nicely analyzes the possibility of SSL approaches to top up the normal contrastive training of CLIP. The twist comes late and it tells that resorting to well-established training recipes reaps more benefit compared to SSL losses. The finding can be useful for typical leaderboard chase as well.
2. The paper does a good job in summarizing and concisely describing the working principles of relevant literature.
3. The experiments are well thought, well designed and well executed. 3 different pretraining datasets and 3 different vision backbones are tested with. The pseudo-codes add to the ease of understanding. Different ablations especially the gradual addition of augmentations and label smoothing variation are nice additions to the main experiments.

Weaknesses:
1. Although the experimental plans are pretty good, a sensitivity analysis of the performance (on imagenet zero-shot accuracy) on $\alpha$ and $\beta$ hyperparameters (ref. eqn. (6)) would have been good to have.

---

> ### Author Response · Authors · 2023-07-10
> **Response to Reviewer Z9sJ**
>
> We are delighted that Reviewer Z9sJ appreciated our systematic investigation of SSL-inspired losses for vision-language pre-training and we would like to express our gratitude for their valuable feedback. To this regard, we addressed the requested changes:
>
> 1. **[Sensitivity analysis wrt. $\alpha$ and $\beta$]** We ran an ablation of the $\alpha$ and $\beta$ hyper-parameters regulating the weights of the contrastive and non-contrastive terms, respectively. For the sake of computation, we carried out the analysis only for BarLIP – the most principled non-contrastive approach –, pre-trained with ResNet-50 on YFCC15M for 32 epochs. We ablate values of $\beta$ while keeping $\alpha$ fixed at 1 and vice versa. Results are reported in Table C.
>
> Table C. Ablation of \alpha and \beta.
>
>
> | $\alpha$ | $\beta$ | YFCC15M |
> |:-----:|:----:|:-------:|
> | 1     | 0    | 43.2    |
> | 1     | 0.25 | 42.8    |
> | 1     | 0.50 | 42.9    |
> | 1     | 0.75 | 42.4    |
> | 0     | 1    | 0.1     |
> | 0.25  | 1    | 38.9    |
> | 0.50  | 1    | 41.6    |
> | 0.75  | 1    |  42.5       |
> | 1     | 1    | 42.6    |
>
> In Table C, we observe that increasing the value of $\beta$ with $\alpha$ fixed at 1 slightly harms the performance of up to 0.8%, while doing the opposite, i.e.,  increasing $\alpha$ with $\beta$ fixed at 1, leads to a substantial drop of accuracy, up to 3.7%. Notably, using only the non-contrastive loss ($\alpha=0$, $\beta=1$) is problematic for zero-shot evaluation because the CLIP projector is not optimized and it is hard to perform zero-shot classification on the non-contrastive projector (see answer to Reviewer WDqT). These results are inline with the claims made in the paper and further confirm the ineffectiveness of adopting non-contrastive losses together with our improved recipe.
>
> 2. [**typos**] We fixed the reported typos; the changes will be reflected in the final PDF:
>     1. SSL acronym is now introduced in the second paragraph of page 1 ''... and self-supervised representation learning (SSL). …''
>     2. Modified as suggested

---

### Decision · Action_Editors · 2023-08-27

**Recommendation:** Accept as is

**Comment:**

The study re-evaluates the leaderboard race, emphasizing the superiority of traditional training over SSL for enhancing CLIP. With a clear summary of related works and robust experiments across multiple datasets, it introduces an efficient training recipe combining image/text augmentations and projector designs. This method surpasses complex CLIP models and shows potential for scalability. Design choices are solidly backed by detailed ablation studies.

Initially, the reviewers raised concerns about in-depth hyperparameter analysis and ablations of self-supervised learning. One reviewer is also expecting a deeper theoretical insight and broader scaling experiments, especially with larger models. After the revision, all the reviewers were satisfied and voted to accept.

A Feature Certification is recommended for this work. Indeed, all the reviewers acknowledged that this paper is a non-trivial deviation from the leaderboard chase. The idea is simple yet effective. Its design philosophy may shed light on future vision-language pretraining research.

**Audience:**

Yes.

**Claims And Evidence:**

Yes, all claims supported by clear presentation and sufficient experimental evaluation.